# Essential Role of Visfatin in Lipopolysaccharide and Colon Ascendens Stent Peritonitis-Induced Acute Lung Injury

**DOI:** 10.3390/ijms20071678

**Published:** 2019-04-04

**Authors:** Yi-Chen Lee, Chun-Yu Lin, Yen-Hsu Chen, Wen-Chin Chiu, Yen-Yun Wang, Chin Hsu, Stephen Chu-Sung Hu, Yu-Han Su, Shyng-Shiou F. Yuan

**Affiliations:** 1Translational Research Center, Kaohsiung Medical University Hospital, Kaohsiung Medical University, Kaohsiung 807, Taiwan; yichen83@kmu.edu.tw (Y.-C.L.); wlkb3065@gmail.com (W.-C.C.); yuhansu0528@gmail.com (Y.-H.S); 2Department of Anatomy, School of Medicine, College of Medicine, Kaohsiung Medical University, Kaohsiung 807, Taiwan; 3School of Medicine, Graduate Institute of Medicine, Sepsis Research Center, Center of Dengue Fever Control and Research, Kaohsiung Medical University, Kaohsiung 807, Taiwan; infectionman@gmail.com (C.-Y.L.); d810070@kmu.edu.tw (Y.-H.C.); 4Division of Infectious Diseases, Department of Internal Medicine, Kaohsiung Medical University Hospital, Kaohsiung Medical University, Kaohsiung 807, Taiwan; 5Department of Internal Medicine, Kaohsiung Municipal Ta-Tung Hospital, Kaohsiung 801, Taiwan; 6Department of Biological Science and Technology, College of Biological Science and Technology, National Chiao Tung University, Hsinchu 300, Taiwan; 7Division of Thoracic Surgery, Department of Surgery, Kaohsiung Medical University Hospital, Kaohsiung Medical University, Kaohsiung 807, Taiwan; 8School of Dentistry, College of Dental Medicine, Kaohsiung Medical University, Kaohsiung 807, Taiwan; winnie30304@hotmail.com; 9Department of Physiology, School of Medicine, College of Medicine, Kaohsiung Medical University, Kaohsiung 807, Taiwan; chinhsu@kmu.edu.tw; 10Department of Dermatology, Kaohsiung Medical University Hospital, Kaohsiung, Medical University, Kaohsiung 807, Taiwan; stephenhu30@hotmail.com; 11Department of Medical Research and Department of Obstetrics and Gynecology, Kaohsiung Medical University Hospital, Kaohsiung Medical University, Kaohsiung 807, Taiwan; 12Graduate Institute of Medicine, Kaohsiung Medical University, Kaohsiung 807, Taiwan; 13Center for Intelligent Drug Systems and Smart Bio-devices (IDS2B), National Chiao Tung University, Hsinchu 300, Taiwan

**Keywords:** acute lung injury, visfatin, FK866

## Abstract

Acute lung injury (ALI) is a life-threatening syndrome characterized by acute and severe hypoxemic respiratory failure. Visfatin, which is known as an obesity-related cytokine with pro-inflammatory activities, plays a role in regulation of inflammatory cytokines. The mechanisms of ALI remain unclear in critically ill patients. Survival in ALI patients appear to be influenced by the stress generated by mechanical ventilation and by ALI-associated factors that initiate the inflammatory response. The objective for this study was to understand the mechanisms of how visfatin regulates inflammatory cytokines and promotes ALI. The expression of visfatin was evaluated in ALI patients and mouse sepsis models. Moreover, the underlying mechanisms were investigated using human bronchial epithelial cell lines, BEAS-2B and NL-20. An increase of serum visfatin was discovered in ALI patients compared to normal controls. Results from hematoxylin and eosin (H&E) and immunohistochemistry staining also showed that visfatin protein was upregulated in mouse sepsis models. Moreover, lipopolysaccharide (LPS) induced visfatin expression, activated the STAT3/NFκB pathway, and increased the expression of pro-inflammatory cytokines, including IL1-β, IL-6, and TNF-α in human bronchial epithelial cell lines NL-20 and BEAS-2B. Co-treatment of visfatin inhibitor FK866 reversed the activation of the STAT3/NFκB pathway and the increase of pro-inflammatory cytokines induced by LPS. Our study provides new evidence for the involvement of visfatin and down-stream events in acute lung injury. Further studies are required to confirm whether the anti-visfatin approaches can improve ALI patient survival by alleviating the pro-inflammatory process.

## 1. Introduction

Acute respiratory distress syndrome (ARDS) is a severe form of acute lung injury (ALI) that is characterized by severe lung tissue inflammation, reduced gas exchange, and, finally, acute hypoxemic respiratory failure [1,2]. In addition, ALI/ARDS is associated with a high mortality rate, between 20% and 50% [3,4,5], and is triggered by various factors such as bacteria and virus infection, major trauma, massive blood transfusion, pneumonia, and sepsis [6,7,8]. Nevertheless, the detailed mechanism of ALI/ARDS remains poorly understood. While the advances in the restrictive fluid management, lung protection strategies, and antibiotics treatment of infection of ALI and ARDS have improved the outcomes, there is still no effective treatment for ALI/ARDS [9,10,11,12]. Lipopolysaccharide (LPS), a component of the outer membrane of bacteria, provokes strong immune reactions in animals through complement activation and has been widely used to study the mechanisms of ALI [13,14]. Colon ascendens stent peritonitis (CASP) procedure, by creating an intestinal leakage of feces and resulting in diffuse peritonitis and polymicrobial sepsis, is also a frequently used mouse model for ALI [13]. Moreover, recent studies showed that ALI/ARDS increased the release of proinflammatory cytokines, such as tumor necrosis factor (TNF-α), interleukin (IL)-1β, IL-6, and IL-8 from immune cells [15,16], and increased the permeability of the alveolar–capillary barrier [8,17]. These phenomena indicate that acute inflammation plays an important role in the ALI/ARDS process. Hence, developing more effective strategies to inhibit inflammatory responses and identifying new diagnostic and therapeutic targets are critical for the improvement of patient outcomes.

Visfatin, also known as pre-B-cell colony-enhancing factor (PBEF) and nicotinamide phosphoribosyltransferase (Nampt), is a 52-kDa molecule first identified as a cytokine involved in B-cell maturation [18]. It is highly expressed in bone marrow, liver, and muscle, and is secreted by a variety of cell types, including lymphocytes, neutrophils, macrophages, and epithelial cells [18,19,20,21]. Visfatin protein has several biologic functions according to its intracellular or extracellular localization [18,22,23]. In the intracellular compartment, visfatin acts as nicotinamide phosphoribosyltransferase (Nampt), which is involved in the mammalian salvage pathway of NAD synthesis [24,25]. Visfatin has been reported as a new inflammatory cytokine that can increase the production of other inflammatory cytokines, such as IL-8, IL-6, IL-1β, and TNF-α in human monocytes [26,27,28]. Furthermore, visfatin is involved in several inflammatory diseases, including psoriasis, rheumatoid arthritis, atherosclerosis, inflammatory bowel diseases, and ALI in sepsis patients [26,29,30,31,32].

A previous study reported that visfatin is highly expressed in ALI of dog, mouse, and human, and two specific visfatin single nucleotide polymorphisms are associated with higher risk of ALI [33]. However, the detailed mechanisms of how visfatin regulates inflammatory cytokines and promotes ALI remain poorly understood. Herein, we report that the elevated visfatin levels led to decreased mouse survival in two ALI mouse models (LPS and CASP), which was rescued by visfatin inhibitor FK866. In addition, in vitro studies show that visfatin stimulates the expression of inflammatory cytokines through STAT3 and NFκB pathways.

## 2. Results

### 2.1. Visfatin Correlates with Poor Survival and Inflammatory Cytokines in ALI Patients

Acute lung injury (ALI), defined as an inflammatory disease in the lung, is mostly caused by over-secretion of pro-inflammatory cytokines [1,34]. It is present with complex autocrine and paracrine inter-relationship of cytokines and pro-inflammatory mediators that initiate and amplify the inflammatory activity. In this study, we first determined the protein levels of visfatin in ALI patients. Between 2013 and 2014, 892 consecutive patients with sepsis were enrolled in the emergency department of Kaohsiung Medical University Hospital. ALI was diagnosed in 110 septic patients. Of these patients, 25 died and 85 survived. Serum visfatin levels were significantly higher in ALI patients, in comparison to normal controls (Figure 1A). Moreover, the survival rate of ALI patients was negatively correlated with visfatin levels (Figure 1B). Since visfatin regulates pro-inflammatory activities through release of inflammatory cytokines, including interleukin1-β (IL1-β, interleukin-6(IL-6), and tumor necrosis factor-α (TNF-α [28,35], we studied the correlation of visfatin with pro-inflammatory cytokines in ALI patients. As shown in Figure 1C, the elevated serum visfatin levels were correlated with IL-6, IL-8, IL-10, and monocyte chemotactic protein-1 (MCP-1) in the ALI patients. 

### 2.2. Visfatin Expression is Upregulated in Both LPS and CASP-Induced ALI Mice, While Visfatin Inhibitor FK866 Attenuated the ALI Phenotypes 

Since serum visfatin was elevated in ALI patients, we further evaluated the in vivo activity of visfatin in mouse models of ALI, including LPS model and CASP model [36,37]. After sacrifice of the mice with LPS or CASP treatment, we performed histological examination by H&E staining of the sections from lung and bronchus tissues, which showed an increase of damage in lung and bronchus tissues, accompanied by septal thickening and interstitial infiltration of neutrophils and lymphocytes (Figure 2A,B). FK866, a visfatin inhibitor, has been indicated as a novel therapeutic compound that attenuates acute lung injury via modulation of innate immune functions [38,39]. In this study, co-treatment of FK866 successfully attenuated the LPS and CASP-induced acute injury in lung and bronchus tissues (Figure 2A,B). 

We also analyzed whether FK866 can rescue mouse death caused by treatment of LPS and CASP. Mice with FK866 co-treatment had higher survival rate than those without FK866 treatment in the LPS model (Figure 2C) and the CASP model (Figure 2D). We further investigated the in vivo regulation of visfatin protein in ALI mouse models. Serum visfatin levels were elevated in both LPS and CASP-treated mice, in comparison to mock-treated mice (Figure 3A,B). Signal transducer and activator of transcription 3 (STAT3) acts as a mediator and biomarker in endothelial activation [40]. Previous studies indicated that visfatin promotes endothelial angiogenesis by activating STAT3, characterized by increased tyrosine phosphorylation of residue 705 (Y705), nuclear translocation, and DNA binding activity in human endothelial cells [41,42,43]. Visfatin was reported to cause endothelial dysfunction by increasing inflammatory and adhesion molecule expression through the upregulation of NF-κB activity [44]. Furthermore, the results of immunohistochemistry demonstrated that LPS and CASP treatment dramatically increased visfatin, and p-STAT3(Y705) and p-NF-κB-p65 protein expression in lung and bronchial tissues, while FK866 co-treatment reversed the elevated visfatin, and p-STAT3(Y705) and p-NF-κB-p65 expression caused by LPS or CASP treatment (Figure 3C–H).

### 2.3. LPS-Upregulated Visfatin Expression in Human Bronchial Epithelial Cells Mediates the Activation of the STAT3/NF-κB Pathway

Given the results that visfatin was elevated in both ALI patients and LPS and CASP mouse models, we further investigated the underlying mechanisms using human bronchial epithelial cell lines BEAS-2B and NL-20. After LPS treatment, visfatin expression increased in dose- and time-dependent manners (Figure 4A,B). 

To gain insight into the signal pathways involved in the regulation of visfatin in ALI, the activity of STAT3 and NF-kB in bronchial epithelial cells was examined. Western blot result showed that, upon LPS treatment, the expression of p-STAT3(Y705) was upregulated in dose- and time-dependent manners (Figure 4A,B). We further tested whether visfatin regulates the NF-κB activity in bronchial epithelial cells. As expected, the expression of S536-phosphorylated/activated p65 (a subunit of NF-κB) was increased upon LPS treatment (Figure 4A,B). Furthermore, the increased visfatin expression upon LPS treatment was correlated with the elevated expression of inflammatory cytokines, including IL-1β, TNF-α, and IL-6 (Figure 4A,B). 

To further confirm that visfatin indeed activates STAT3 and NF-κB, we co-treated FK866 with LPS in BEAS-2B. Co-treatment of FK866, which decreased visfatin expression, also led to a decrease in activation of STAT3 and NF-κB, as well as a decrease in their downstream effectors, including IL1-β, IL-6, and TNF-α (Figure 4C). Next, we tested whether STAT3 regulates NF-κB activity by co-treatment of static, a STAT3 inhibitor, with LPS. As shown in Figure 4D, co-treatment of static and LPS decreased the expression of p-NFκB-p65 and downstream inflammatory cytokines, including IL1-β, IL-6, and TNF-α (Figure 4D). Together, these results support that LPS upregulates visfatin expression and leads to an activated STAT3/NF-κB pathway and down-stream inflammatory cytokine, like IL1-β, IL-6, and TNF-α.

## 3. Discussion

The pathophysiological mechanisms of acute lung injury are complex and not well understood yet. In this study, we applied LPS and CASP mouse models and a LPS cell model to address the molecular mechanisms behind ALI. Our data clearly demonstrated that visfatin induction activated the STAT3/NFκB pathway and downstream inflammatory cytokines in LPS and CASP-induced ALI. Besides, we provide evidence that inhibition of visfatin activity by a pharmacologic approach (FK866) reverses the inflammatory process caused by LPS and CASP, and improves survival in LPS and CASP mouse models. A schematic representation for LPS-induced signaling pathway in human bronchial epithelial cells was shown in Figure 4E.

In this study, we observed that serum visfatin levels were positively correlated with the serum levels of pro-inflammatory cytokines IL-6, IL-8, IL-10, and MCP-1, not IL1-β or TNF-α, in ALI patients. However, LPS treatment promoted the expression of visfatin and pro-inflammatory cytokines IL1-β, IL-6, and TNF-α in human bronchial epithelial cell lines (Figure 4A,B). While we do not have the answer for this discrepancy in the patterns of pro-inflammatory cytokines in vivo and in vitro, it might be caused by the difference between the complex clinical manifestations in ALI and the over-simplified LPS and CASP models. Another possibility is the involvement of other pathways, than visfatin, in the induction of pro-inflammatory cytokines in ALI patients. For example, NF-κB transcriptional activities and inflammatory lung injury are induced by visfatin via its direct binding to Toll–like receptor 4 (TLR4) and; therefore, visfatin can serve as an innate immunity molecule capable of activating TLR4 in the absence of LPS, which delineates a novel dimension to the induction of lung inflammatory responses by non-infectious mechanisms [45].

In our bronchial epithelial cell model for LPS, we observed the sequential changes in visfatin elevation, STAT/NF-κB activation, and the increase of pro-inflammatory cytokines, which were reversed by co-treatment of FK866 (Figure 4). Moreover, FK866 treatment alleviated the inflammatory changes in the lung tissues and improved the survival in the LPS- and CASP-treated mice (Figure 2 and Figure 3). A recent report also showed that FK866 treatment protects intestinal integrity and reduces bacteremia via the NF-κB-mediated pathway [38]. A NAMPT inhibitor analog, meta-carborane-butyl-3-(3-pyridinyl)-2E-propenamide (MC-PPEA, MC4), was synthesized by replacing the benzoylpiperidine moiety in FK866 with meta-carborane [46]. MC4 also showed inhibitory effects on cecal ligation and puncture (CLP)-induced mortality, TNFα levels, pulmonary myeloperoxidase activity, alveolar injury, and IL-6 and IL-1β messenger RNA levels [47]. Taken together, we discovered that LPS or CASP induced visfatin protein expression and ensured STAT3 or NF-κB activation and induced downstream inflammatory cytokines expression, and this was further confirmed in the cell model (Figure 4).

In conclusion, our study provides in vitro and in vivo evidence for the involvement of visfatin and down-stream events in the pathogenic changes in acute lung injury. The clinical application of FK866 in ALI patients, especially those with elevated visfatin locally and systemically, should be explored to confirm whether the anti-visfatin approaches can improve the inflammatory changes and survival in ALI patients.

## 4. Materials and Methods

### 4.1. Patient Enrollment

This study was approved by the Institutional Review Board of Kaohsiung Medical University Hospital. Patients with sepsis and visiting the Emergency Department of Kaohsiung Medical University Hospital were sequentially enrolled between 2013 and 2014. Patients received treatments and examinations according to emergency physicians’ evaluation. After the patients agreed and provided informed consent, 10 mL of blood were collected for cytokine analysis. Serum was separated by centrifugation at 1600× *g* for 20 min at 4 °C, divided in aliquots and immediately frozen (−80 °C) until further analysis. Clinical information including the patients’ age, gender, microbiological culture results, laboratory data at the Emergency Department, and diagnosis were evaluated by clinicians and recorded by a research nurse. Patients meeting criteria for both sepsis [48] and ALI [49] were enrolled. Thirty-two adult healthy persons, who came to the hospital for their routine healthy examination, were enrolled as a control group.

### 4.2. Reagents

Antibody against p-Stat-3(pS727)(sc-8001) was purchased from Santa Cruz Biotechnology, Inc. (Dallas, TX, USA). Antibodies against p-Stat-3(pY705)(#9145), p-NFκB-p65(pS536)(#8242), and NFκB -p65(#3033) were purchased from Cell Signaling Technology, Inc. (Danvers, MA, USA). Antibodies against Stat-3(GTX15523), IL1-β(GTX22105), TNF-α(GTX110520), and IL-6 (GTX110527) were purchased from GeneTex, Inc. (Irvine, CA, USA). Antibody against visfatin(ab58640) was purchased from Abcam plc. (Cambridge, UK). Antibody against β-actin(A5411) was purchased from Sigma-Aldrich Co. (St. Louis, MO, USA). LHC-9 medium was purchased from Invitrogen (Carlsbad, CA, USA). FK866, lipopolysaccharide, hematoxylin, and eosin were purchased from Sigma-Aldrich (St Louis, MO, USA).

### 4.3. Cell Culture

The human bronchial epithelial cell lines, NL-20 (BCRC) and BEAS-2B (ATCC), were cultured in LHC-9 medium (Invitrogen, NY, USA) supplemented with 5% fetal bovine serum, 100 U/mL penicillin, and 100 pg/mL streptomycin (Invitrogen) in a humidified incubator with 5% CO_2_ at 37 °C.

### 4.4. Animal Model of Abdominal Sepsis

Eight-week-old BALB/C mice were purchased from Lasco, Taiwan. The technique used for induction of colon ascendens stent peritonitis (CASP) was performed as reported recently [50]. A 16-gauge venous catheter was prepared by creating a notch at a distance of 2 mm from the orifice. One millimeter beyond, the catheter was cut following laparotomy. The colon ascendens was exteriorized and a 7/0 ethilon thread (Ethicon, Nordersted, Germany) was stitched through the antimesenteric portion of the colon ascendens, about 10 mm distal to the ileocecal valve. The 16-gauge venous catheter was punctured antimesenterically through the colonic wall into the intestinal lumen, directly proximal of a pre-tied knot, fixed, and the second puncture was performed directly proximal of the stent, again strictly antimesenterically. Then, the inner needle was removed and the stent was cut at the prepared site. To ensure proper intraluminal positioning of the stent, stool was milked from the cecum into the colon ascendens, until a small drop appeared (1 mm in diameter). Fluid resuscitation of animals was performed by flushing 0.5 mL sterile saline solution into the peritoneal cavity before closure of the abdominal wall. After operation, mice were intraperitoneally injected with or without 20 mg/kg of FK866 in 100 µL normal saline every 3 h. 

For the LPS experiment, 20 mg/kg of LPS and/or 20 mg/kg of FK866 were injected intraperitoneally. The animal study was approved by the Institutional Animal Care and Use Committee (IACUC No.104056) of Kaohsiung Medical University.

### 4.5. Immunoblotting Analysis

The whole cell extracts were collected by lysing cells in the buffer containing 50 mM HEPES, 6 mM EDTA, and 1% Triton X-100 supplemented with phosphatase inhibitor (Sigma, St. Louis, MO, USA) and complete protease inhibitor cocktail (Roche Applied Science, Indianapolis, IN, USA). Total protein was separated by centrifugation at 12000× *g* for 20 min at 4 °C. The protein concentration was determined using the Bradford assay (Bio-Rad, Hercules, CA, USA), with bovine serum albumin as a standard. Equal amounts of proteins were subjected to 10% sodium dodecyl sulfate polyacrylamide gel electrophoresis and transferred onto nitrocellulose membranes (Pall Corporation, East Hills, NY, USA). The chemiluminescent signal was captured by a ChemiDocTM XRS+ System (Bio-Rad Laboratories, Hercules, CA, USA) and quantified with Image Lab software (Bio-Rad Laboratories, Hercules, CA, USA).

### 4.6. Enzyme Immunoassay

Serum visfatin levels for mouse and human were measured in duplicate by a visfatin-specific enzyme immunoassay kit, RayBiotech (Norcross, GA, USA) and BioVendor (Brno, Czech Republic), respectively, according to the manufacturer’s instructions.

### 4.7. Immunohistochemistry 

Immunohistochemical staining for visfatin and IL-6 was performed on an automated Bond-Max instrument (Leica Microsystems, Wetzlar, Germany). Slides carrying tissue sections cut from formalin-fixed and paraffin-embedded tissue blocks were dried for 1 h at 60 °C. These slides were then covered by Bond Universal Covertiles and placed into the Bond-Max instrument. All subsequent steps were performed by the automated instrument as follows: (1) Deparaffinization of tissues on the slides by rinsing with Bond Dewax Solution at 72 °C; (2) heat-induced epitope retrieval (antigen unmasking) with Bond Epitope Retrieval Solution 1 for 20 min at 100 °C; (3) peroxide block placement on the slides for 5 min at room temperature; (4) incubation with rabbit antibody at a dilution of 1:200 for 30 min at room temperature; (5) Bond Polymer placement on the slides for 8 min at room temperature; (6) color development with DAB (3,3’-diaminobenzidine tetrahydrochloride) as a chromogen for 5 min at room temperature; and (7) hematoxylin counter-staining for 5 min, followed by mounting of the slides and then examination by light microscopy. The pictures were captured by a Nikon E-800M microscope (Tokyo, Japan). For quantification, the staining of visfatin, p-STAT, and NFκB was scored using the method of histochemical score (H-score) [51], which was calculated as the product of percentage of stained cells and intensity of staining

### 4.8. Statistical Analysis

Data were given as mean ± SD and all experiments were repeated at least three times independently. All statistical analyses were performed using the SPSS 20.0 statistical software for PC (SPSS, Chicago, IL, USA) and *p* values less than 0.05 were considered statistically significant. Statistical analysis between the different groups was performed using two-sided Student’s *t*-test. Spearman’s correlation coefficient was used to determine the relationship between variables. One-way ANOVA, combined with Tukey’s multiple-comparison test, was used to evaluate the statistical significance of differences between three or more groups.

## Figures and Tables

**Figure 1 ijms-20-01678-f001:**
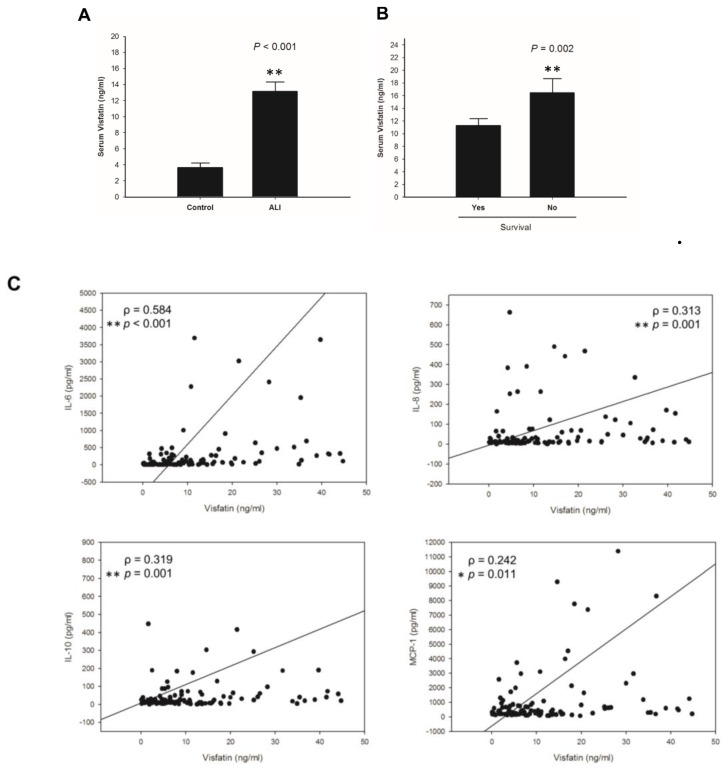
Upregulation of serum visfatin in acute lung injury (ALI) patients. (**A**) Serum visfatin levels in ALI patients (*n* = 110) and normal controls (*n* = 32). (**B**) Correlation of patient survival with serum visfatin levels. (**C**) Spearman’s correlation of serum visfatin with IL-6, IL-8, IL-10, and MCP-1 in ALI patients. Results are mean ± SD. *, *p* value < 0.05. **, *p* value < 0.01.

**Figure 2 ijms-20-01678-f002:**
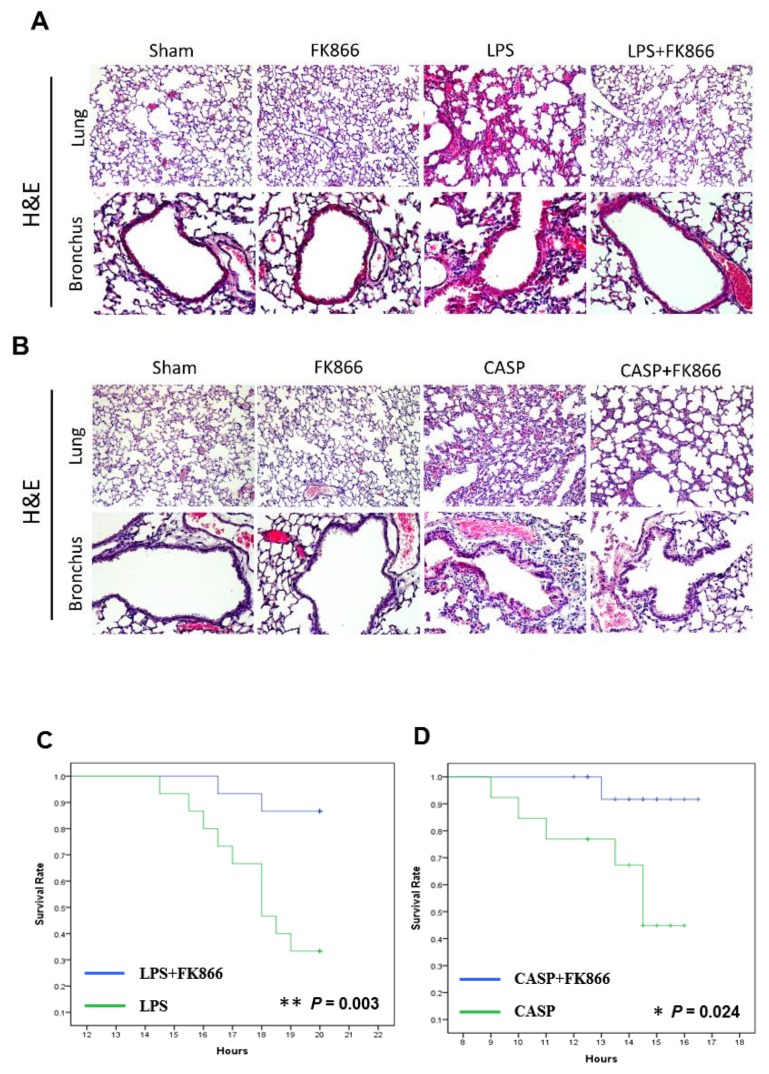
FK866, a visfatin inhibitor, decreased acute lung injury and increased survival in LPS and CASP ALI mouse models. (**A**,**B**) Histology of lung tissues in mice treated with LPS alone, CASP alone, or co-treatment with FK866, determined by H&E staining. Sham mice show normal lung tissue and LPS- or CASP-treated mice show injuries in lung tissue, including fluid-filled alveoli, septal thickening, and interstitial lymphocytic and neutrophilic infiltration. (100× amplification). (**C**,**D**) Survival rate in mice treated with LPS alone, CASP alone (*n* = 15), or co-treatment with FK866 (*n* = 15) in ALI mouse model. *, *p* value < 0.05. **, *p* value < 0.01.

**Figure 3 ijms-20-01678-f003:**
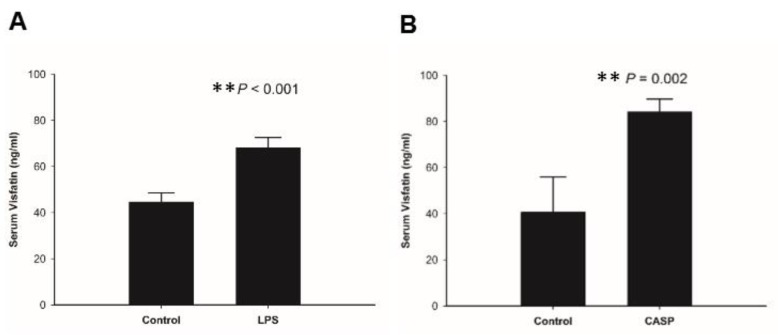
LPS and CASP treatment in mice enhanced visfatin expression in circulation and in lung tissues. (**A**,**B**) Serum visfatin levels in sham mice and LPS- and CASP-treated mice. (**C**,**D**) Visfatin, (**E**,**F**) p-STAT3(Y705), and (**G**,**H**) p-NF-κB-p65 expression in lung tissues in sham mice and LPS- and CASP-treated mice, determined by immunohistochemistry. The representative results are from three separate experiments. Results are mean ± SD (*n* = 3). *, *p* value < 0.05. **, *p* value < 0.01. (100× amplification).

**Figure 4 ijms-20-01678-f004:**
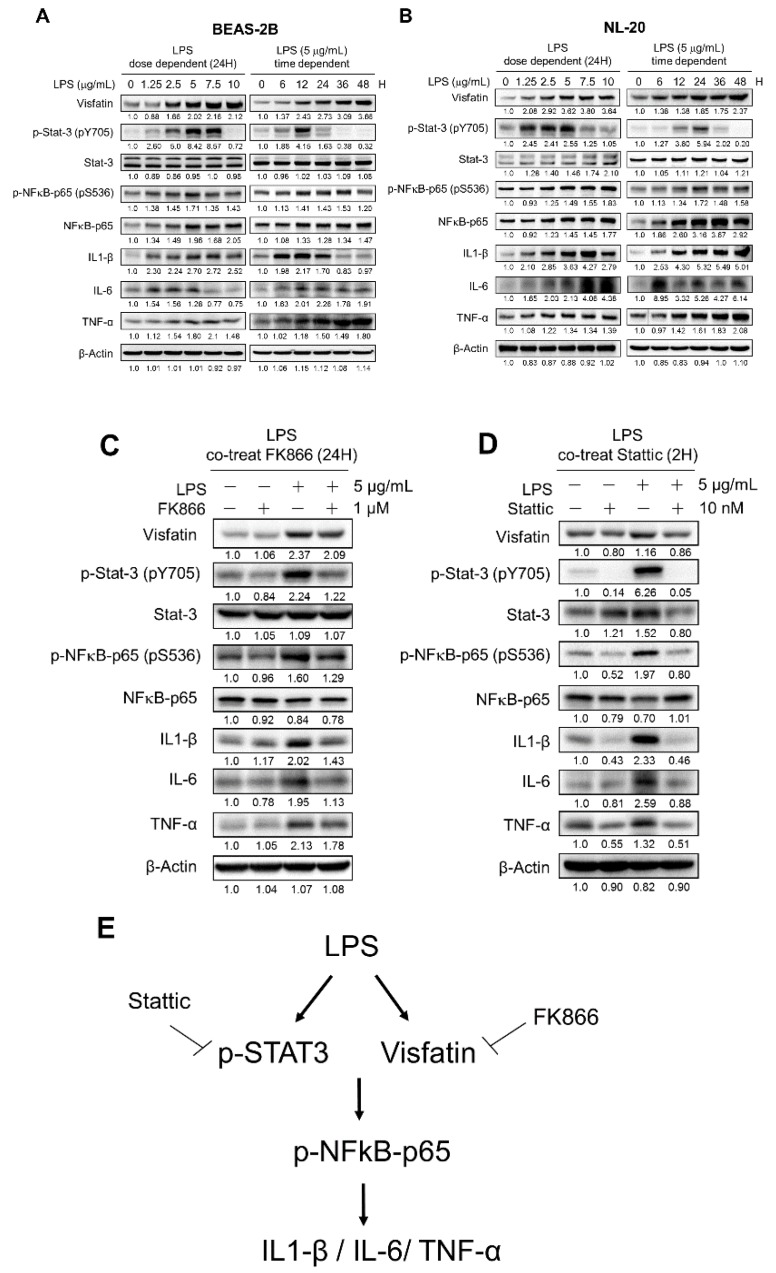
Visfatin increased the expression of pro-inflammatory cytokines through STAT/NF-κB pathway in human bronchial epithelial cells. (**A**,**B**) Dose- and time-dependent effects of LPS on the expression of visfatin, STAT3, NFκB, and pro-inflammatory cytokines in human bronchial epithelial cell lines BEAS-2B and NL-20. (**C**) Co-treatment of FK866 and LPS on the expression of visfatin, STAT3, NFκB, and pro-inflammatory cytokines. (**D**) Co-treatment of STAT3 inhibitor static and LPS on the expression of visfatin, STAT3, NFκB, and pro-inflammatory cytokines. (**E**) Schematic representation of the LPS-induced signaling pathway in human bronchial epithelial cells.

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
