# Peer review of "Essential Role of Visfatin in Lipopolysaccharide and Colon Ascendens Stent Peritonitis-Induced Acute Lung Injury"

_ijms, 2019, doi:10.3390/ijms20071678_

Round 1
Reviewer 1 Report
Interesting manuscript, presenting both in vivo and in vitro data on the role of visfatin in ALI/sepsis.
Only some minor comments:
1) Fig. 1A. Please correct legend figure to ALI (instead of sepsis).
2) Please update and use the Sepsis-3 definitions.
3) Fig. 4A-D. Please also show the densitometries
4) Please correct throughout mean+/-sd (not means).
5) Fig.4E correct stat(t)ic
Author Response
Response to Reviewer 1 Comments
1) Fig. 1A. Please correct legend figure to ALI (instead of sepsis).
Response: We thank the Reviewer for the correction and the legend is corrected in Fig. 1A (P6)
2) Please update and use the Sepsis-3 definitions.
Response: The Sepsis-3 definitions is updated in Materials and Methods (reference 48, P13L413)
3) Fig. 4A-D. Please also show the densitometries
Response: We thank the Reviewer for the suggestion. The densitometries are added in Fig. 4A-D (P11).
4) Please correct throughout mean+/-sd (not means).
Response: We thank the Reviewer for the correction and the word is rephrased in manuscript (P6L246, P10L295 and P14L486).
5) Fig.4E correct stat(t)ic
Response: We thank the Reviewer for the correction and the word is rephrased in manuscript (P11L352)

Reviewer 2 Report
Lee et al (or Yuan et al) have detailed a study on the role of PBEF in septic ALI, with a translational component in human samples. This is a nice bedside to bench study, with some mechanistic implications, as well as human relevance. There are a few issues before I can recommend acceptance.
MAJOR:
1) Figure 1A shows higher PBEF in ARDS patients, relative to controls, but wouldn't the appropriate control be septic non-ARDS subjects? What are the PBEF levels in the 782 septic patients who did not have ARDS, or did the authors not collect blood on these subjects?
2) Figure 1C is not at all convincing, likely because of the non-normal distribution of the cytokines. Please either log-transform and then check correlation, or use the non-parametric spearman's rho.
3) PBEF can activate TLR4 by itself (Sci Rep. 2015 Aug 14;5:13135). This should be discussed, and the findings re-interpreted in light of these data.
MINOR:
1) The first authors are listed differently on the submission website (Yuan) versus the manuscript (Lee). Please make the same.
2) CASP needs to be defined in the Introduction. Right now, it is just an abbreviation without definition until later defined in the Results.
Author Response
Response to Reviewer 2 Comments
MAJOR:
1) Figure 1A shows higher PBEF in ARDS patients, relative to controls, but wouldn't the appropriate control be septic non-ARDS subjects? What are the PBEF levels in the 782 septic patients who did not have ARDS, or did the authors not collect blood on these subjects?
Response: We thank the Reviewer’s comment. The controls in Figure 1A were adult healthy persons (not septic non-ARDS subjects) who came to the hospital for their routine healthy examination. We had provided the information in Materials and Methods (P13L414-415). We did not detect the PBEF levels in the 782 septic patients who did not have ARDS. However, we agree this is an important question and worthy of further studies.
2) Figure 1C is not at all convincing, likely because of the non-normal distribution of the cytokines. Please either log-transform and then check correlation, or use the non-parametric spearman's rho.
Response: We thank the Reviewer’s correction. The correlation of PBEF with the cytokines in Figure 1C is changed to spearman’s correlation method (P6L245) and a description about spearman’s correlation is added in Materials and Methods (P14L489-490).
3) PBEF can activate TLR4 by itself (Sci Rep. 2015 Aug 14;5:13135). This should be discussed, and the findings re-interpreted in light of these data.
Response: We thank the Reviewer’s comment and a sentence is added in Discussion (P12L373-377), Reference (P18L653-656) and Abbreviations (P15L508).
MINOR:
1) The first authors are listed differently on the submission website (Yuan) versus the manuscript (Lee). Please make the same.
Response: We thank the Reviewer’ correction. The authorship(P1L5-6) is correct in the manuscript and we will revise it in MDPI Submission System
2) CASP needs to be defined in the Introduction. Right now, it is just an abbreviation without definition until later defined in the Results.
Response: We thank the Reviewer’s comment and the sentence for CASP definition is moved to Introduction (P3L94-99).
